## PERSPECTIVE

# Improving skeletal muscle insulin sensitivity via beta₂-agonist administration: a promising strategy to counteract metabolic disease and muscle loss

**Marlou L. Dirks** 

*Department of Sport and Health Sciences, College of Life and Environmental Sciences, University of Exeter, Exeter, UK*

Email: m.dirks@exeter.ac.uk

Edited by: Kim Barrett & Bettina Mittendorfer

Linked articles: This Perspective article highlights an article by Jessen *et al*. To read this paper, visit https://doi.org/10.1113/JP282421.

The peer review history is available in the Supporting information section of this article (https://doi.org/10.1113/JP282992#support-information-section).

## The Journal of Physiology

Skeletal muscle tissue is important for functional capacity and activities of daily living, but is also a substantial regulator of metabolic health. As the loss of muscle mass and development of insulin resistance (i.e. impaired insulin-stimulated glucose uptake) negatively affect our healthspan, there is ongoing focus on designing effective interventional strategies to preserve muscle mass and metabolic health in catabolic situations. Such interventions can include nutritional, exercise and pharmacological approaches, although optimising nutrition and performing (resistance-type) exercise are often challenging in vulnerable populations. As part of the development of potent pharmacological interventions, beta₂-adrenoceptor agonists ('beta₂-agonists') are a specific group of drugs that has received attention over the years. As first-line drugs to induce bronchodilatation in obstructive respiratory disease, beta₂-agonists are currently among the most widely used drugs worldwide. Interestingly, a large proportion of the administered drug reaches the systemic circulation, where it has wider effects on skeletal muscle and adipose tissue. As an example, they exert a considerable hypertrophic effect on skeletal muscle, demonstrated by a ~1 kg increase in lean mass over a 4-week treatment period (Jessen et al., 2018). However, although the insulin-sensitising effect of chronic treatment with beta₂-agonists is demonstrated in animals, the translation to human models has been limited.

In this issue of *The Journal of Physiology*, Jessen and co-workers publish novel work using chronic beta₂-agonist administration at therapeutic doses to improve insulin sensitivity in healthy young men (Jessen et al., 2022). The authors conducted a thoroughly controlled human intervention study to assess the impact of daily inhalation of the selective beta₂-agonist terbutaline on insulin-stimulated whole-body glucose disposal and potentially relevant physiological mechanisms in both skeletal muscle and adipose tissue. Prior to and following the 4-week intervention, whole-body glucose disposal was measured via gold-standard hyperinsulinaemic–euglycaemic clamps, and whole-body and regional lean and fat mass via dual-energy X-ray absorptiometry. Strikingly, the authors demonstrated a substantial ~25% increase in glucose infusion rate following 4 weeks of terbutaline inhalation. Importantly, due to a rigorous 48−72 h washout period following the final dose of the experimental drug, this effect can be attributed to the chronic administration rather than an acute effect of the final dose(s). The authors thereby corroborate previous work, in which an almost fourfold greater dose of terbutaline sulphate was given and a comparable 29% increase in insulin-stimulated glucose disposal was observed (Scheidegger et al., 1984). The finding that doses in a close-to-therapeutic range induce a similar insulin-sensitising effect as higher doses likely aids in minimising the potential side effects (e.g. tremor and palpitations) of beta₂-agonist administration. The improvement in insulin sensitivity was accompanied by a favourable change in body composition, characterised by a 1.1 kg increase in lean mass and concomitant 0.5 kg fat mass loss. Skeletal muscle is the major site for insulin-stimulated glucose uptake, with >80% of glucose disposal occurring in muscle under these hyperinsulinaemic–euglycaemic conditions (Jessen et al., 2022). Indeed, the authors demonstrate that the observed changes in lean mass are strongly correlated with the improvements in insulin-stimulated glucose uptake, suggesting that the positive effects of chronic beta₂-agonist administration on insulin sensitivity are, potentially in large part, attributable to muscle hypertrophy. Despite this tight correlation, other changes in skeletal muscle tissue (e.g. preferential oxidation of glucose over fat and glycogen accumulation; Scheidegger et al., 1984) likely explain part of the increase in glucose disposal. Although Jessen and colleagues did not observe changes in protein content of GLUT4 or hexokinase in muscle tissue (Jessen et al., 2022), previous work has proposed that these effects are (at least in part) mediated via activation of a novel beta₂-adrenoceptor signalling pathway (Sato et al., 2014). This pathway leads to GLUT4 translocation via mammalian target of rapamycin complex 2 phosphorylation, largely independently from the phosphoinositide 3-kinase–Akt signalling pathway. Unfortunately, the current study design does not provide further insight in the metabolic fate of the infused glucose. Despite this, it is thought that the potent insulin-sensitising effects of beta₂-agonists are mediated both by an increase in the amount of muscle tissue (i.e. increased capacity) and by an enhancement of muscle glucose uptake per unit of muscle (i.e. increased efficiency).

Although the majority of research on the effects and mechanisms of beta₂-agonist treatment is conducted in animals, due to a growing ageing population and increased prevalence of chronic metabolic disease there is an urgency to develop effective interventional strategies to maintain muscle mass and metabolic health throughout the lifespan. Because of the demonstrated hypertrophic and insulin-sensitising effects, beta₂-agonists treatment is a promising strategy in conditions characterised by muscle atrophy and insulin resistance. To illustrate the potency of chronic beta₂-agonist treatment, the measured 25% difference between the terbutaline and placebo groups (Jessen et al., 2022) is of similar magnitude to the difference between glucose-tolerant individuals and individuals with type 2 diabetes, and equivalent to the decline observed following 30–40 years of ageing, and may therefore be able to (partly) overcome the observed metabolic

deterioration in these conditions. Moreover, given the concomitant stimulatory effect on both muscle mass and insulin sensitivity, the administration of short-acting beta$_2$-agonists during periods of limb immobilisation or bed rest is a highly promising approach to attenuate or even prevent muscle disuse-induced atrophy and insulin resistance (Dirks et al., 2020). In summary, chronic beta$_2$-agonist treatment is a highly promising interventional strategy to improve muscle glucose uptake, with potential to translate these exciting findings to conditions characterised by muscle loss and insulin resistance.

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

## Additional information

### Competing interests

None.

### Author contributions

Sole author.

### Funding

None.

### Keywords

glucose uptake, insulin resistance, muscle atrophy, skeletal muscle

## Supporting information

Additional supporting information can be found online in the Supporting Information section at the end of the HTML view of the article. Supporting information files available:

**Peer Review History**

