## [Peer Review History · The Journal of Physiology]

Improving skeletal muscle insulin sensitivity via beta2-agonist administration: a promising strategy to counteract metabolic disease and muscle loss

Marlou L. Dirks

DOI: 10.1113/JP282992

Corresponding author(s): Marlou Dirks (m.dirks@exeter.ac.uk)

Review Timeline:

Submission Date:

25-Mar-2022

Accepted:

11-Apr-2022

Senior Editor: Kim Barrett

Reviewing Editor: Bettina Mittendorfer

Transaction Report:

Dear Dr Dirks,

Re: JP-P-2022-282992 "Improving skeletal muscle insulin sensitivity via beta2-agonist administration: a promising strategy to counteract metabolic disease and muscle loss" by Marlou L. Dirks

I am pleased to tell you that your invited Perspective article has been accepted for publication in The Journal of Physiology.

NEW POLICY: In order to improve the transparency of its peer review process The Journal of Physiology publishes online as supporting information the peer review history of all articles accepted for publication. Readers will have access to decision letters, including all Editors' comments and referee reports, for each version of the manuscript and any author responses to peer review comments. Referees can decide whether or not they wish to be named on the peer review history document.

The last Word version of the paper submitted will be used by the Production Editors to prepare your proof. When this is ready you will receive an email containing a link to Wiley's Online Proofing System. The proof should be checked and corrected as quickly as possible.

All queries at proof stage should be sent to tjp@wiley.com.

Thank you very much for your contribution to The Journal of Physiology.

Yours sincerely,

Professor Kim E. Barrett
Editor-in-Chief
The Journal of Physiology
<https://jp.msubmit.net>
<http://jp.physoc.org>
The Physiological Society
Hodgkin Huxley House
30 Farringdon Lane
London, EC1R 3AW
UK
<http://www.physoc.org>
<http://journals.physoc.org>

Reviewing Editor Comments:

Thank you for the nice contribution.

Reviewer Comments:

Dr. Dirks provides an interesting perspective on our research regarding the augmented insulin-stimulated glucose with inhaled beta2-agonist. The Perspectives is well-written and factually correct.

Confidential Review**25-Mar-2022**